# Ferroptosis-related genes mediate tumor microenvironment and prognosis in triple-negative breast cancer via integrated RNA-seq analysis

Xuantong Gong[1†], Lishuang Gu[2†], Di Yang[1], Yu He[1], Qian Li[3*], Hao Qin[4*], Yong Wang[1,5*]

[1]Department of Ultrasound, National Cancer Center/National Clinical Research Center for Cancer/Cancer Hospital, Chinese Academy of Medical Sciences and Peking Union Medical College, Beijing, China; [2]Department of Ultrasound, Beijing Hospital of Traditional Chinese Medicine, Capital Medical University, Beijing, China; [3]Department of Ultrasound, Affiliated Cancer Hospital of Zhengzhou University, Zhengzhou, China; [4]Key Laboratory of Cancer and Microbiome, State Key Laboratory of Molecular Oncology, National Cancer Center/National Clinical Research Center for Cancer/Cancer Hospital, Chinese Academy of Medical Sciences and Peking Union Medical College, Beijing, China; [5]Department of Ultrasound, The First Affiliated Hospital of China Medical University, Shenyang, Shenyang, China

*For correspondence:
754427296@qq.com (QL);
haoqin@cicams.ac.cn (HQ);
drwangyong77@163.com (YW)

[†]These authors contributed equally to this work

## eLife Assessment

This study presents a **useful** finding for the ferroptosis-mediated tumor microenvironment (TME) in triple-negative breast cancer (TNBC) using public single-cell RNA sequencing (scRNA-seq) and bulk RNA sequencing data. The data were collected and analyzed using **solid** and validated methodology and can be used as a starting point for functional studies of TME in TNBC. The work will be of interest to medical biologists working in the field of TNBC.

**Abstract** Triple-negative breast cancer (TNBC), an aggressive malignancy with limited tools to predict recurrence and drug sensitivity, exhibits ferroptotic heterogeneity across subtypes. However, the tumor microenvironment (TME) mediated by ferroptosis-related genes remains poorly characterized. This study integrates single-cell and bulk RNA sequencing data from the Gene Expression Omnibus to elucidate ferroptosis-driven TME features in TNBC, employing machine learning to develop prognostic and therapeutic response prediction models. At the single-cell level, T cells were classified into three subpopulations and macrophages into two subpopulations, with their infiltration degrees significantly correlated with clinical outcomes. A risk score model constructed based on these findings demonstrated robust predictive performance, validated in external cohorts with 3-, 4-, and 5-year area under the receiver operating characteristic curves of 0.65, 0.67, and 0.71, respectively. Notably, high-risk patients exhibited enhanced sensitivity to 27 therapeutic agents. By delineating ferroptosis-associated immune heterogeneity, this work provides a risk stratification tool to enhance prognostic precision and therapeutic decision-making in TNBC, while identifying genes offer actionable targets for TNBC precision medicine.

## Introduction

Global cancer statistics show that breast cancer is the most common cancer in women and the leading cause of cancer deaths (*Bray et al., 2024*). Triple-negative breast cancer (TNBC) is a subtype of breast cancer that lacks expression of estrogen receptor, progesterone receptor, and human epidermal growth factor receptor-2. Compared with other subtypes, TNBC is highly aggressive, with a poor overall prognosis for patients and a median survival of less than 1 year after recurrent metastasis (*Bardia et al., 2019*; *Howlader et al., 2018*; *Keenan and Tolaney, 2020*; *Yam et al., 2021*). Due to the lack of targets for endocrine therapy and targeted therapeutic agents, chemotherapy and emerging immunotherapy are the main therapies. TNBC is highly heterogeneous and only some patients benefit from treatment (*Cortazar et al., 2014*; *Kim et al., 2018*; *Shepherd et al., 2022*). Unfortunately, there is no effective method that can effectively predict the prognostic risk and drug sensitivity of TNBC, which is an urgent problem in the clinic.

The tumor microenvironment (TME) is composed of various cell types such as cancer-associated fibroblasts, tumor-associated macrophages, T cells, NK cells, B cells, and endothelial cells, which interact with cancer cells and influence various aspects such as tumor progression, metastasis, and response to therapy (*Binnewies et al., 2018*; *Mao et al., 2021*; *Mayer et al., 2023*). The composition and functional status of TME vary greatly among different patients with breast cancer, and revealing the TME of each patient is crucial for selecting a reasonable treatment and controlling tumor progression in the long term (*de Visser and Joyce, 2023*; *Yang et al., 2023b*).

Ferroptosis is an iron-dependent form of non-apoptotic, oxidative form of regulated cell death involving lipid hydroperoxides (*Fang et al., 2023*; *Viswanathan et al., 2017*). Studies have shown that drug-resistant cancer cells are more sensitive to ferroptosis (*Hangauer et al., 2017*; *Kim et al., 2022*; *Tsoi et al., 2018*). Therefore, ferroptosis is more recognized as a potential target for cancer therapy. Ferroptotic heterogeneity exists in different subtypes of TNBC (*Yang et al., 2023a*). Since ferroptosis is regulated by multiple metabolic pathways, the ferroptosis landscape of TNBC remains unexplored and its relationship with patient prognosis and treatment response is uncertain (*Lu et al., 2022*). Therefore, exploring ferroptosis-related genes in TNBC and their correlation with the immune microenvironment may provide a new treatment trend for TNBC.

RNA sequencing (RNA-seq) technology is a gene expression analysis method that can qualitatively and quantitatively explore the transcriptome characteristics of biological samples at the tissue and cellular levels (*Hong et al., 2020*; *Kuksin et al., 2021*). Single-cell RNA-seq (scRNA-seq) has greatly enhanced our understanding of transcriptional heterogeneity across cell types and states, and can be used to explore the TME (*Li et al., 2024*; *Ma et al., 2023*). Bulk RNA-seq is used to analyze the average expression levels of RNA in tissues or cell populations, and can be utilized to explore differences among individuals (*Chen et al., 2022*; *Janjic et al., 2022*). Machine learning is a branch of artificial intelligence that enables computer systems to learn from training data and gain experience, and it is increasingly being used for the diagnosis, treatment, and prognostic evaluation of cancer (*Ching et al., 2018*; *Guan et al., 2024*). Literature reports that several models have been constructed based on RNA-seq data for the prognostic evaluation of breast cancer patients, but the reproducibility and interpretability of the results still deserve further exploration (*Gao et al., 2024*; *Lei et al., 2022*; *Pei et al., 2023*; *Zhao et al., 2024*). This study intends to further explore the relationship among ferroptosis genes, prognosis, and drug sensitivity from the perspective of ferroptotic heterogeneity in TNBC. In this work, we collected and integrated data from single-cell and bulk RNA-seq to characterize the ferroptosis-related gene mediated TME landscape of TNBC on multiple scales. Furthermore, based on the findings in TME, predictive models of patient prognosis and treatment response were constructed using machine learning algorithms. We hope to provide a new way of individualized risk management for triple-negative breast cancer patients and assist their precision treatment.

## Results

### The single-cell landscape of TNBC samples

In this study, a total of nine TNBC single-cell samples were included. After quality control (QC), 38,985 genes from 29,733 cells per sample were finally selected for subsequent analysis (*Figure 1—figure supplement 1*). Based on original annotations of data, these cells were divided into 9 major cell clusters and 29 minor cell clusters (*Figure 1a–c*). Major cell clusters include B-cells, cancer-associated

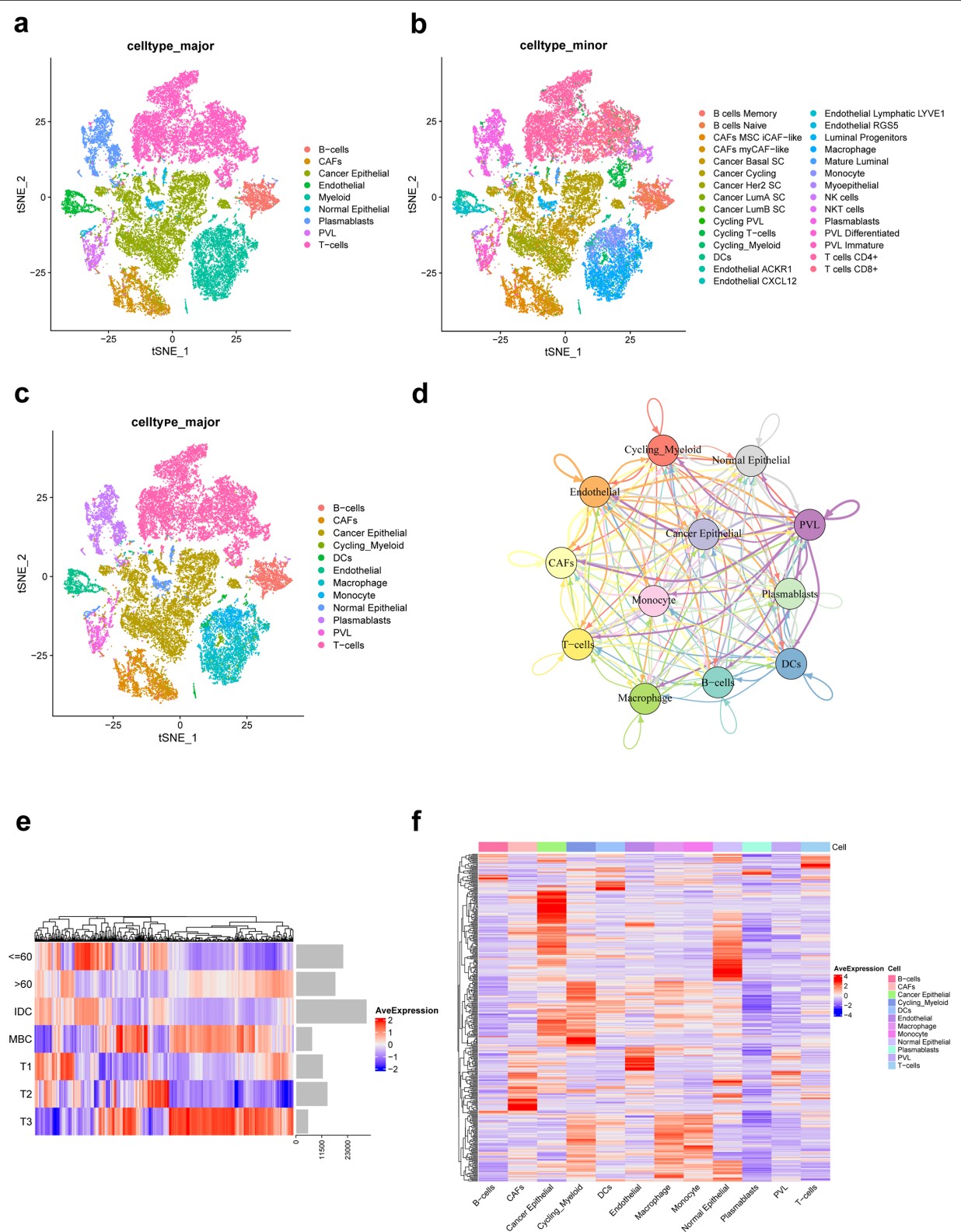

**Figure 1.** Integration and clustering of scRNA-seq data from triple-negative breast cancer. (**a**) t-SNE plot of the nine major cell clusters. (**b**) t-SNE plot of the 29 minor cell clusters. (**c**) t-SNE plot of the 12 major cell clusters. (**d**) The number of ligand–receptor interactions of different major cell clusters in cellcell communication network, different colors represent different cell clusters, and arrows represent ligandreceptor orientation. (**e**) Heat map showing

*Figure 1 continued on next page*

*Figure 1 continued*

the average expression of ferroptosis-related genes in different clinicopathological classifications. (**f**) Heat map showing the average expression of ferroptosis-related genes in different major cell clusters.

The online version of this article includes the following source code and figure supplement(s) for figure 1:

**Source code 1.** Single-cell RNA-seq data preprocessing in triple-negative breast cancer.

**Figure supplement 1.** Single-cell RNA-seq data preprocessing in triple-negative breast cancer.

fibroblasts (CAFs), cancer epithelial, endothelial, myeloid, normal epithelial, plasmablasts, perivascular-like cells (PVL), T-cells. Of these, myeloid cluster is further divided into cycling_myeloid, dendritic cells (DCs), macrophage, and monocyte clusters. As shown in *Figure 1c*, the immune microenvironment of TNBC is characterized by a high proportion of T cells and macrophages. On this basis, ligand–receptor interactions between different major cell clusters are frequent (*Figure 1d*).

Of the 471 ferroptosis-related genes from the FerrDb website, 391 were present in the single-cell expression matrix of TNBC. We found that there were significant differences in the average expression of these genes in different clinicopathological classifications, as shown in *Figure 1e*. Moreover, there were significant differences in the expression of ferroptosis-related genes in 12 major cell clusters of TNBC (*Figure 1f*).

## Ferroptosis-related subpopulations of T cells in the TNBC

A total of 11,784 T cells in the scRNA-seq data of this study, including cycling T cells, NK cells, NKT cells, T cells CD4+, and T cells CD8+, identified clusters (*Figure 2a*). Based on the non-negative matrix factorization (NMF) algorithm (rank = 3), each T cell cluster was further categorized into three ferroptosis-related subpopulations (T_C1, T_C2, T_C3). *Figure 2b* shows that differentially expressed top 50 ferroptosis-related genes differed significantly among these three subpopulations. Among them, the proportion of NK cells in the T_C2 subpopulation was significantly more than the other two subpopulations (*Figure 2c*). According to pseudotime trajectories, we found that all three T cell subpopulations were involved at different periods of differentiation (*Figure 2d–f*). Further, *Figure 2g* shows the average expression of signature genes associated with 8 functions in 15 subpopulations of T-cells, and we found that gene expression was significantly higher in the cycling T-cells. Moreover, different T-cell subpopulations play an important role in the antitumor process (*Figure 2h*).

Transcription factors (TFs) are key regulators in cellular signal transduction (*Gong et al., 2022b*). As a result, RFX5, EOMES, TBX21, CEBPB, RUNX3, and IKZF3 showed higher activities in NKT cells. Besides, IRF, NFATC2, STAT, and PRDM1 showed higher activities in CD8[+] T cells (*Figure 2i*). The HALLMARK analysis results revealed a prominent enrichment in pathways such as pathways in DNA–Repair and adipogenesis in cycling T cells, whereas in other clusters of T cells, pathways such as epithelial–mesenchymal transition and KRAS signaling had higher enrichment. However, the difference in enrichment scores between ferroptosis-related subpopulations of T cells was not significant.

## Ferroptosis-related subpopulations of macrophages in the TNBC

A total of 3671 macrophages in the scRNA-seq data of this study. Based on the NMF algorithm (rank = 4), two ferroptosis-related subpopulations of macrophages (M_C1, M_C2) were finally obtained after clustering cell clusters with similar markers, and the t-distributed stochastic neighbor embedding (t-SNE) plot is shown in *Figure 3a*. The pseudotime trajectories showed that M_C2 cells belong to the early stage of macrophages and then gradually develop into M_C1 cells (*Figure 3b and c*). *Figure 3d* shows that differentially expressed top 50 ferroptosis-related genes differed significantly among subpopulations.

Further, we found that FOLR2, SEPP1, MRC1, LYVE1, SLC40A1, and CD163 were highly expressed in M_C1 cells and named M_C1 as FOLR2 + MAC. whereas TREM2, FN1, CXCR4, C3, S100A8, IFI6, and SPP1 were highly expressed in M_C2 and named M_C2 as TREM2 + MAC. *Figure 3e and f* shows the marker genes in each subpopulation of macrophages. Enrichment scores of the two subpopulations in each cell are shown by t-SNE plots (*Figure 3g and h*). Moreover, the enrichment score of TREM2 + MAC and FOLR2 + MAC were significantly negatively correlated ($r=-0.779$) (*Figure 3i*). Communication between these two subpopulations and cancer epithelial cells is vigorous (*Figure 3j*).

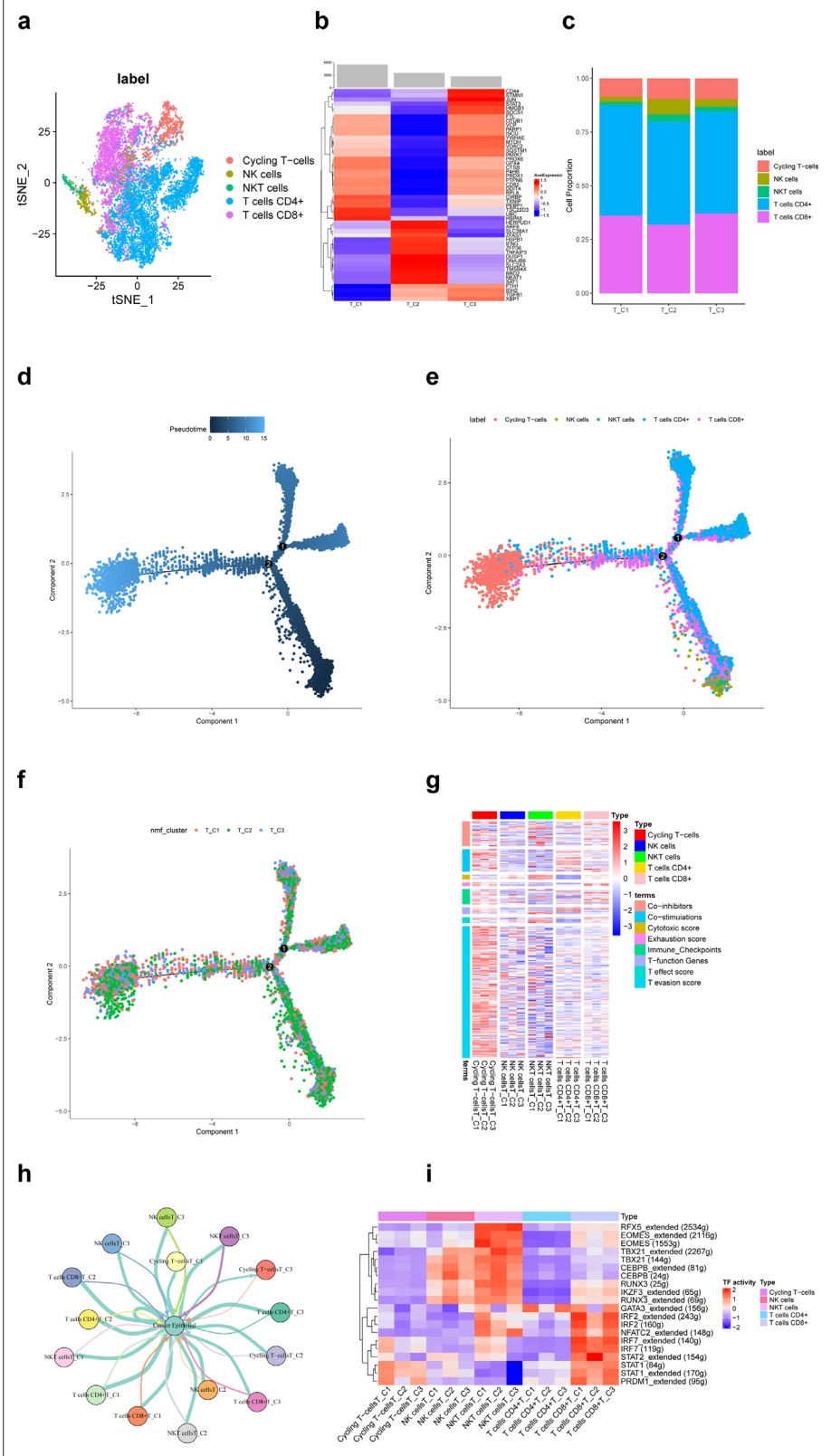

**Figure 2.** Ferroptosis-related subpopulations of T cells in triple-negative breast cancer. (**a**) t-SNE plot of the five identified clusters of T cells. (**b**) Heat map showing the expression of ferroptosis-related genes (top 50) in three subpopulations of T cells. (**c**) Demonstration of the proportion of clusters of T cells in ferroptosis-related subpopulations. (**d**) Pseudotime trajectories showing the developmental time course of T cells. (**e**) Pseudotime

*Figure 2 continued on next page*

*Figure 2 continued*

trajectories of the five identified clusters of T cells. (**f**) Pseudotime trajectories of ferroptosis-related subpopulations of T cells. (**g**) Heat map showing the average expression of signature genes associated with 8 functions in 15 subpopulations of T-cells. (**h**) Network diagram of the cell–cell communication between cancer epithelial cells and T cells. (**i**) Heat map for differential analysis of transcription factors activity.

The online version of this article includes the following source code for figure 2:

**Source code 1.** Ferroptosis-related subpopulations of T cells in triple-negative breast cancer.

In addition, detection of TFs activity in each macrophage cell revealed that most of TFs had higher activity in FOLR2 + MAC cells (*Figure 3—figure supplement 1*).

## Construction and validation of predictive survival models based on ferroptosis-related genes

Based on the recurrence-free survival (RFS) data of TNBC in the GSE25066 database, univariate Cox regression analysis was performed on 8371 specific markers of ferroptosis-related subpopulations, and 41 genes significantly associated with RFS were screened (p<0.01) (*Supplementary file 1*, *Figure 4—figure supplement 1*). These genes were further analyzed by Least Absolute Shrinkage and Selection Operator (LASSO) regression, and 23 genes were finally selected for the construction of the risk model. We calculate the risk score using the following formula: risk score = TMEM160 * (–0.4689) + EWSR1 * (–0.4237) + BCAT2 * (–0.2346) + PNKP * (–0.1592) + MLEC * (–0.1567) + SAP30BP * (–0.1469) + TBL1XR1 * (–0.0894) + STAG1 * (–0.0695) + NR4A1 * (–0.0630) + TNFRSF9 + (–0.0503) + PSD3 * (–0.0153) + BAD * (–0.0094) + MIIP * 0.0104 + HIST3H2A * 0.0294 + CDC25B * 0.0528 + TCEB1 * 0.0670 + HMGCS1 * 0.0988 + SPC25 * 0.1317 + TKT * 0.2234 + PTTG1 * 0.2960 + ADA * 0.3097 + AK1 * 0.4003 + AIMP2 * 0.4081 (*Figure 4a–c*). All patients were categorized into low- and high-risk groups according to the median value of the risk score (*Figure 4d–f*). Survival curves showed that patients in the high-risk group had a shorter RFS compared to patients in the low-risk group (p<0.05, *Figure 4g*). In addition, the risk score had an excellent predictive effect in predicting RFS in TNBC patients, its area under the receiver operating characteristic curves (AUCs) for 3, 4, and 5 years RFS were 0.87, 0.88, and 0.88 (*Figure 4h*).

In external validation set (GSE86166), patients in the high-risk group also had a shorter RFS compared to patients in the low-risk group (p<0.05). This risk model also performed well in predicting RFS in TNBC patients, its AUCs for 3-, 4-, and 5-year RFS were 0.65, 0.67, and 0.71 (*Figure 4i*).

## Analysis of independent prognostic factors

Further, we performed univariate and multivariate Cox analyses to determine whether the risk score could serve as an independent prognostic factor for TNBC patients compared to other common clinicopathologic factors. In the GSE25066 database, *Figure 5a* shows that both the risk factor and stage were significantly associated with TNBC patients' RFS and were independent prognostic factors (p<0.05). In the external validation set, *Figure 5b* shows that only the risk factor can be considered as independent prognostic factor in TNBC patients (p<0.05).

## Correlation of the risk score with immune microenvironment

Next, we calculated the ImmuneScore, StromalScore, ESTIMATEScore, and TumorPurity for samples in the low- and high-risk groups and compared the differences in scores between the two groups. The results showed that these scores did not differ significantly between the low- and high-risk groups (*Figure 6a–f*). Then, based on the CIBERSORT algorithm, we found that T cells CD4 memory activated, NK cells resting, and monocytes had significantly higher proportions in the high-risk group, and the NK cells activated, and mast cells resting had significantly higher proportions in the low-risk group (*Figure 6g*). Infiltration of these immune cells may play an important role in the clinical course of TNBC patients.

## Analysis of clinical response to drugs with the risk score

Based on the 'pRRophetic' R package, we explored the relationship between the risk score and clinical response to 138 drugs and calculated the 50% inhibitory concentration (IC$_{50}$). As a result, 16 of 50

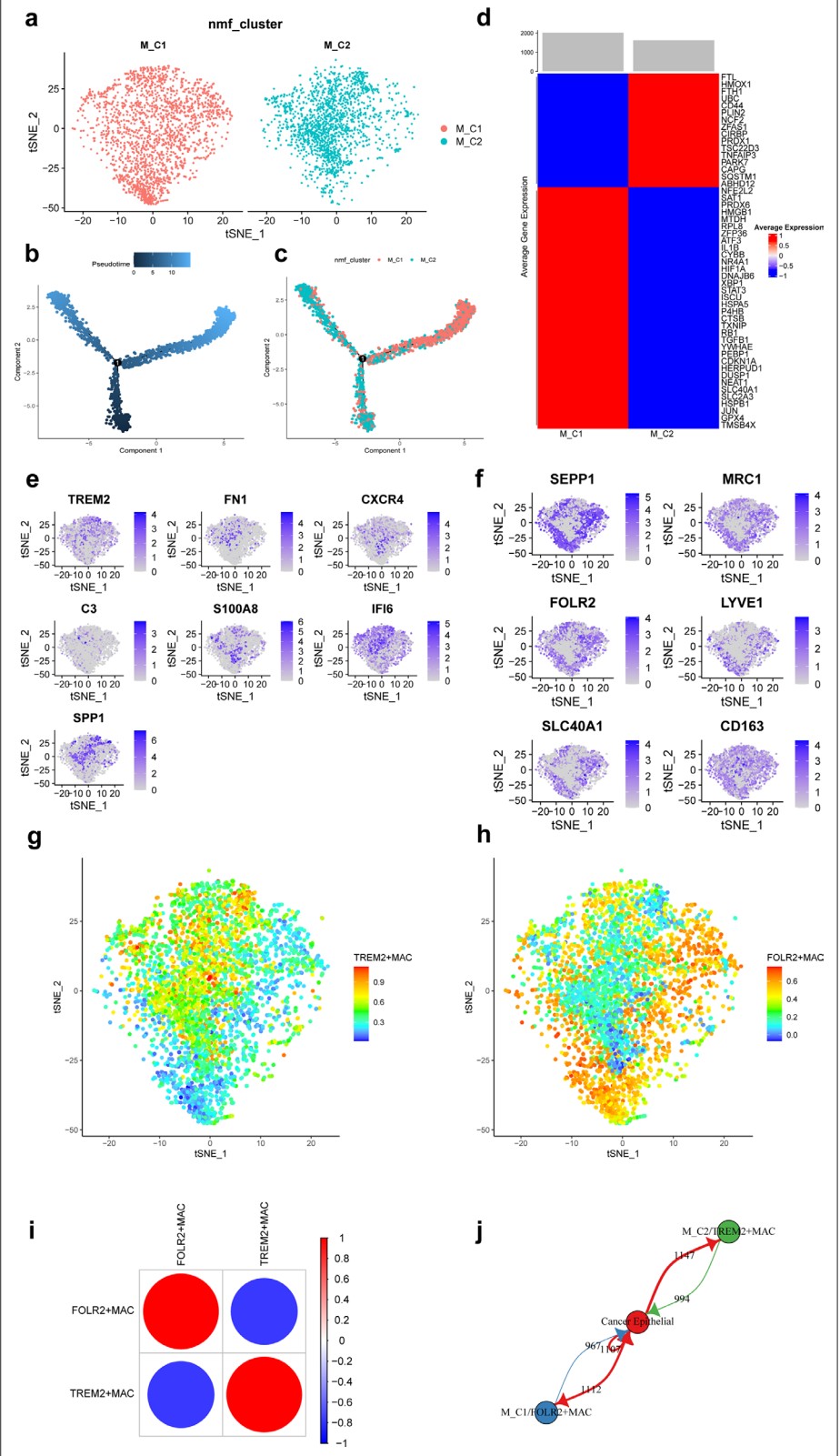

**Figure 3.** Ferroptosis-related subpopulations of macrophages in triple-negative breast cancer. (**a**) t-SNE plot of two ferroptosis-related subpopulations of macrophages. (**b**) Pseudotime trajectories showing the developmental time course of macrophages. (**c**) Pseudotime trajectories of two ferroptosis-related subpopulations of macrophages. (**d**) Heat map showing the average expression of ferroptosis-related genes in two subpopulations of

*Figure 3 continued*

macrophages. (**e**) t-SNE plot showing the expression patterns of marker genes in M_C2. (**f**) t-SNE plot showing the expression patterns of marker genes in M_C1. (**g**) t-SNE plot showing the enrichment score of the TREM2 + MAC in each cell. (**h**) t-SNE plot showing the enrichment score of the FOLR2 + MAC in each cell. (**i**) Bubble plot showing the correlation of enrichment score between TREM2 + MAC and FOLR2 + MAC. (**j**) Network diagram of the cell-cell communication between cancer epithelial cells and ferroptosis-related subpopulations of macrophages.

The online version of this article includes the following source code and figure supplement(s) for figure 3:

**Source code 1.** Ferroptosis-related subpopulations of macrophages in triple-negative breast cancer.

**Figure supplement 1.** Heat map for differential analysis of transcription factors activity among different subpopulations of macrophages in the triple-negative breast cancer.

drugs whose clinical response was significantly associated with the risk score were positively associated with the risk score, and the remaining 34 were negatively associated with the risk score. Further, we found that 27 of drugs negatively associated with the risk score had significantly lower $IC_{50}$ in the high-risk group, suggesting that patients in the high-risk group may be more sensitive to these drugs, favoring the choice of clinical medication (*Figure 7a*). Thirteen drugs positively associated with the risk score had significantly higher $IC_{50}$ in the high-risk group than in the low-risk group, suggesting that patients in the high-risk group may be less sensitive to these drugs (*Figure 7b*).

Finally, we calculated the tumor immune dysfunction and exclusion (TIDE) scores of patients in the low- and high-risk groups; unfortunately, the TIDE scores were not significantly different between the two groups (*Figure 7c*).

## Discussion

TNBC is the most difficult subtype of breast cancer to treat, and survival and efficacy prediction for each individual is a critical step in precision tumor therapy. Several studies have identified ferroptosis-related genes as novel therapeutic targets to enhance treatment efficacy and improve patient prognosis (*Desterke et al., 2023*; *Li et al., 2022*; *Zhang et al., 2021*; *Zhou et al., 2024*). However, to the best of our knowledge, there is a lack of reports on the association of ferroptosis-related genes with prognosis and clinical response to treatment in TNBC. Here, firstly, we revealed ferroptosis-related immune cell clustering in TNBC patients at the single-cell level, demonstrating the ferroptosis heterogeneity in TNBC, which was also supported by other cohorts (*Yang et al., 2023a*). On the basis of the above results, we constructed an ferroptosis-related risk score model based on bulk RNA-seq data, which successfully predicted the RFS of TNBC patients and validated the robustness of the results in various datasets. This is important for the individualized management of TNBC patients. More importantly, the model can also predict the sensitivity to drugs for TNBC patients with different risk stratification to guide physicians in the selection of drugs for different patients, and it has significant clinical application value.

Understanding the TME in TNBC is essential for improving prognosis and guiding treatment. scRNA-seq enables transcriptomic analysis of individual cells to characterize the cellular diversity in the TME in detail, thereby improving understanding of the disease (*Li et al., 2023*; *Van de Sande et al., 2023*; *Zhai et al., 2023*). In this study, our analysis of scRNA-seq data from TNBC revealed that there are multiple immune cells infiltrating in the TME and interacting frequently with each other, and they may impede or promote tumor progression (*Deepak et al., 2020*). We found that the average expression of ferroptosis-related genes was significantly different in different T-stages of tumors, which might be strongly correlated with patient prognosis. Further, based on ferroptosis-related genes, a higher proportion of T cells and macrophages in the TME were categorized into different subpopulations to reveal the heterogeneity of ferroptosis-related in TNBC. The T_C2 subpopulation was accompanied by a greater infiltration of NK cells. NK cells have rapid and efficient antitumor immunity, and their activity is negatively correlated with breast cancer progression (*Bouzidi et al., 2021*; *Gong et al., 2022a*; *Guillerey et al., 2016*). The relevant results were validated in the dataset GSE25066. TFs are universal regulators of the transcription of many genes and are closely associated with the progression of cancer (*Wang et al., 2023*). The results indicate that TBX21, which exhibits higher activity in NKT cells, is associated with the expression of SLC7A11, and overexpression of SLC7A11 alters the expression of chemokines, resulting in the infiltration of immune cells such as CD8[+]

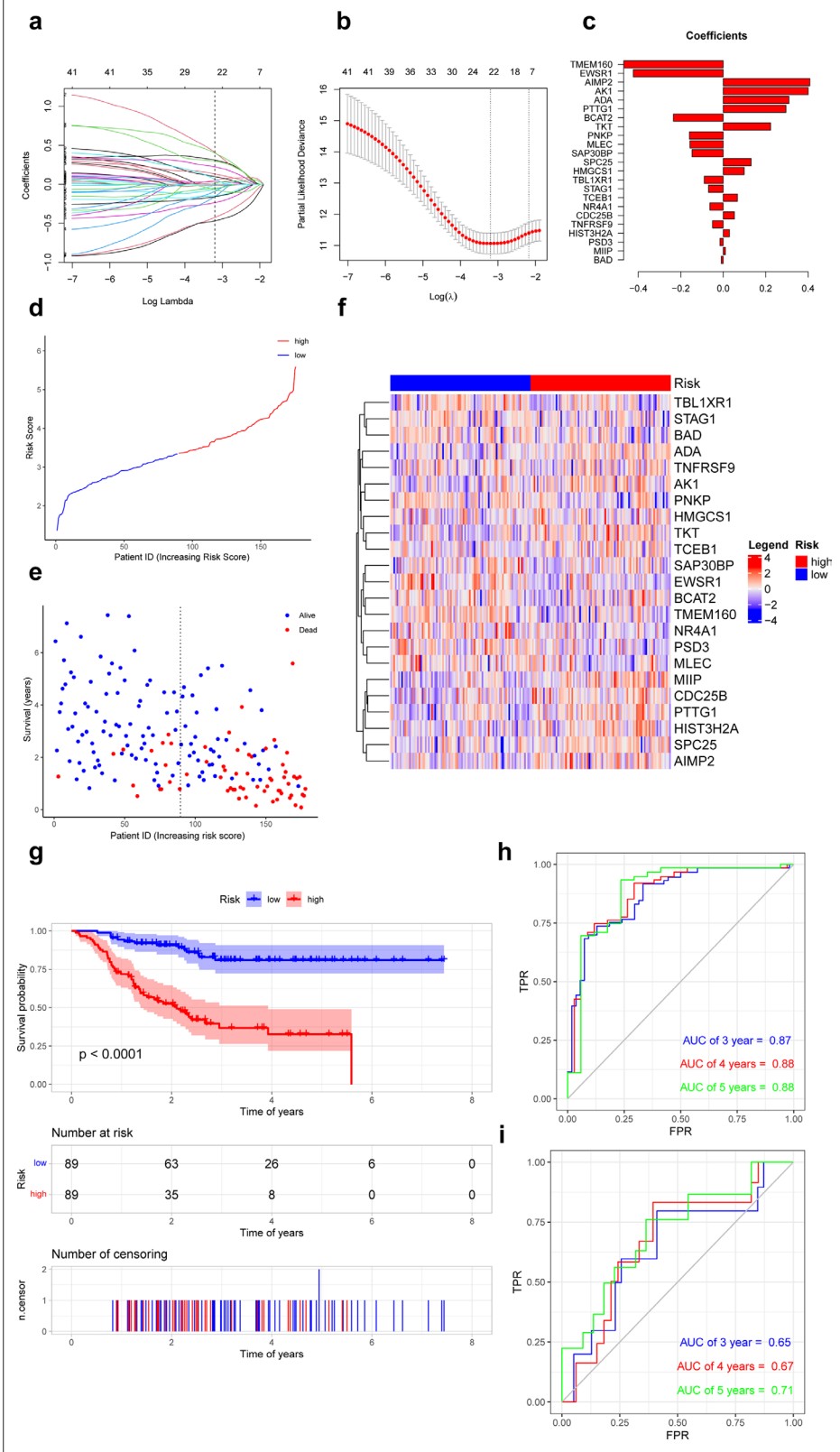

**Figure 4.** Prognostic differences in triple-negative breast cancer with ferroptosis-related subpopulations. (**a**) Least Absolute Shrinkage and Selection Operator (LASSO) regression of 23 ferroptosis-related genes. (**b**) Cross-validation for optimizing the parameter in LASSO regression. (**c**) Demonstration of regression coefficients corresponding to 23 genes. (**d**) Graph showing risk scores for all samples. (**e**) Scatterplot showing recurrence-free

*Figure 4 continued on next page*

*Figure 4 continued*

survival for all samples. (**f**) Heat map showing the average expression of 23 genes in the low- and high-risk groups. (**g**) Kaplan–Meier curves of survival analysis in the low- and high-risk groups. (**h**) Receiver operating characteristic curves for predicting the recurrence-free survival at 3, 4, and 5 years in training set. (**i**) Receiver operating characteristic curves for predicting the recurrence-free survival at 3, 4 and 5 years in external validation set.

The online version of this article includes the following source code and figure supplement(s) for figure 4:

**Source code 1.** Prognostic differences in triple-negative breast cancer with ferroptosis-related subpopulations.

**Figure supplement 1.** The forest plot of the top 20 ferroptosis-related genes significantly associated with disease-free survival based on univariate Cox regression analysis (p0.01).

T cells and neutrophils (*Cheng et al., 2022*). It has been reported that RUNX3 induces ferroptosis by activating the ING1/p53/SLC7A11 signaling pathway, thereby inhibiting the growth of gallbladder cancer (*Cai et al., 2023*). Furthermore, PRDM1 and IRF, which exhibit higher activity in CD8[+] T cells, induce ferroptosis in cancer cells by inhibiting the transcription of GPX4 and driving the expression of transferrin receptor, respectively, thereby improving the TME (*Song et al., 2024*; *Wu et al., 2024*).

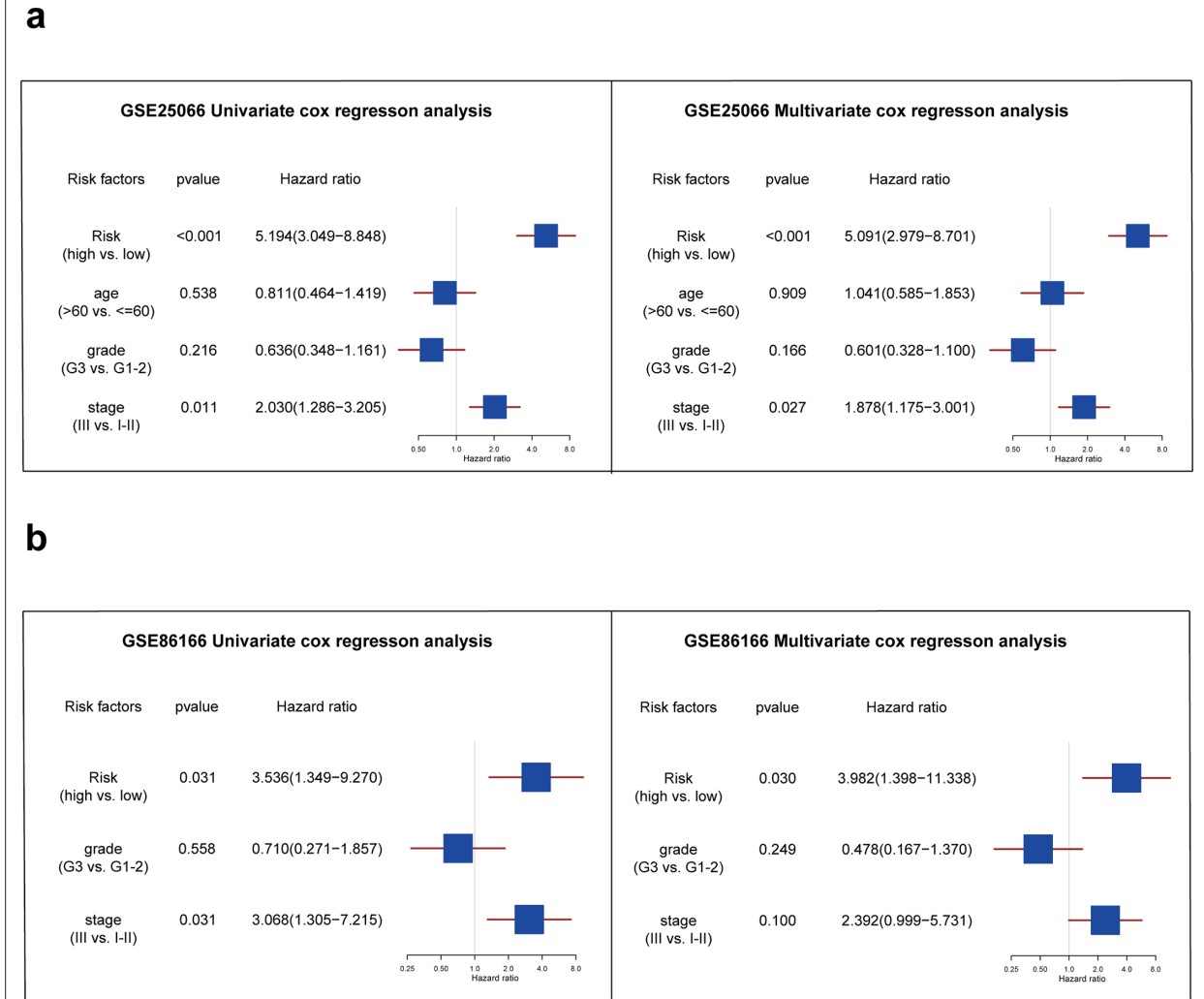

**Figure 5.** Analysis of independent prognostic factors for triple-negative breast cancer patients. (**a**) The forest plot showing the results of univariate and multivariate COX regression analysis of risk score, age, grade, and stage in the GSE25066 database. (**b**) The forest plot showing the results of univariate and multivariate COX regression analysis of risk score, grade, and stage in the GSE86166 database.

The online version of this article includes the following source code for figure 5:

**Source code 1.** Analysis of independent prognostic factors for triple-negative breast cancer patients.

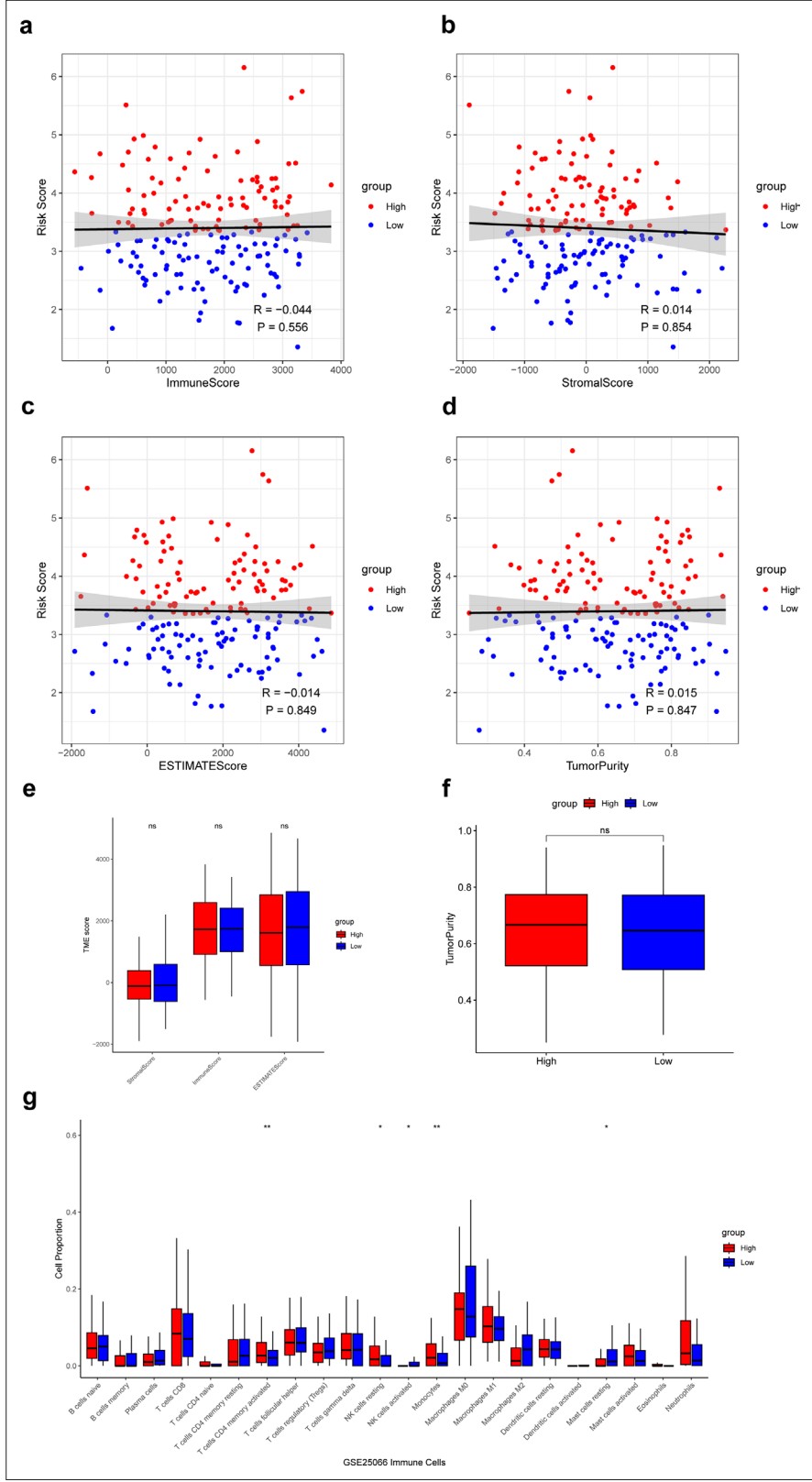

**Figure 6.** Analysis of immune microenvironment in low- and high-risk groups. (**a–d**) Correlation of the risk score with the ImmuneScore, StromalScore, ESTIMATEScore, and TumorPurity. (**e**) Difference of ImmuneScore, StromalScore, and ESTIMATEScore in low- and high-risk groups. (**f**) Difference of TumorPurity in low- and high-risk

*Figure 6 continued on next page*

*Figure 6 continued*

groups. (**g**) Difference of immune infiltration score between low- and high-risk groups calculated by CIBERSORT. *p<0.05, **p<0.01, ns: p>0.05.

The online version of this article includes the following source code for figure 6:

**Source code 1.** Analysis of immune microenvironment in low- and high-risk groups.

FOLR2, SEPP1, MRC1, LYVE1, SLC40A1, and CD163 were highly expressed in the M_C1 subpopulation. Studies have shown that these markers, which are normally present in mammary macrophages from healthy humans and mice, correlate with a favorable prognosis in breast cancer (*Jäppinen et al., 2019*; *Nalio Ramos et al., 2022*; *Wang et al., 2020*). Among them, FOLR2 is positively correlated with antitumor immune players of CD8[+] T cells, DCs, B cells, and tertiary lymphoid structures, and there is a strong correlation between FOLR2 expression and various immune pathways, including T-cell receptor and PD-1 signaling, as well as antigen processing. FOLR2[+] macrophages are an important component in the initiation of antitumor immunity (*Nalio Ramos et al., 2022*). In the M_C2 subpopulation, TREM2, FN1, C3, CXCR4, and SPP1 are highly expressed. It has been shown that these markers are lowly expressed in macrophages of healthy breast tissues and highly expressed in tumor tissues (*Molgora et al., 2020*; *Müller et al., 2001*; *Nalio Ramos et al., 2022*). Specifically, the absence of

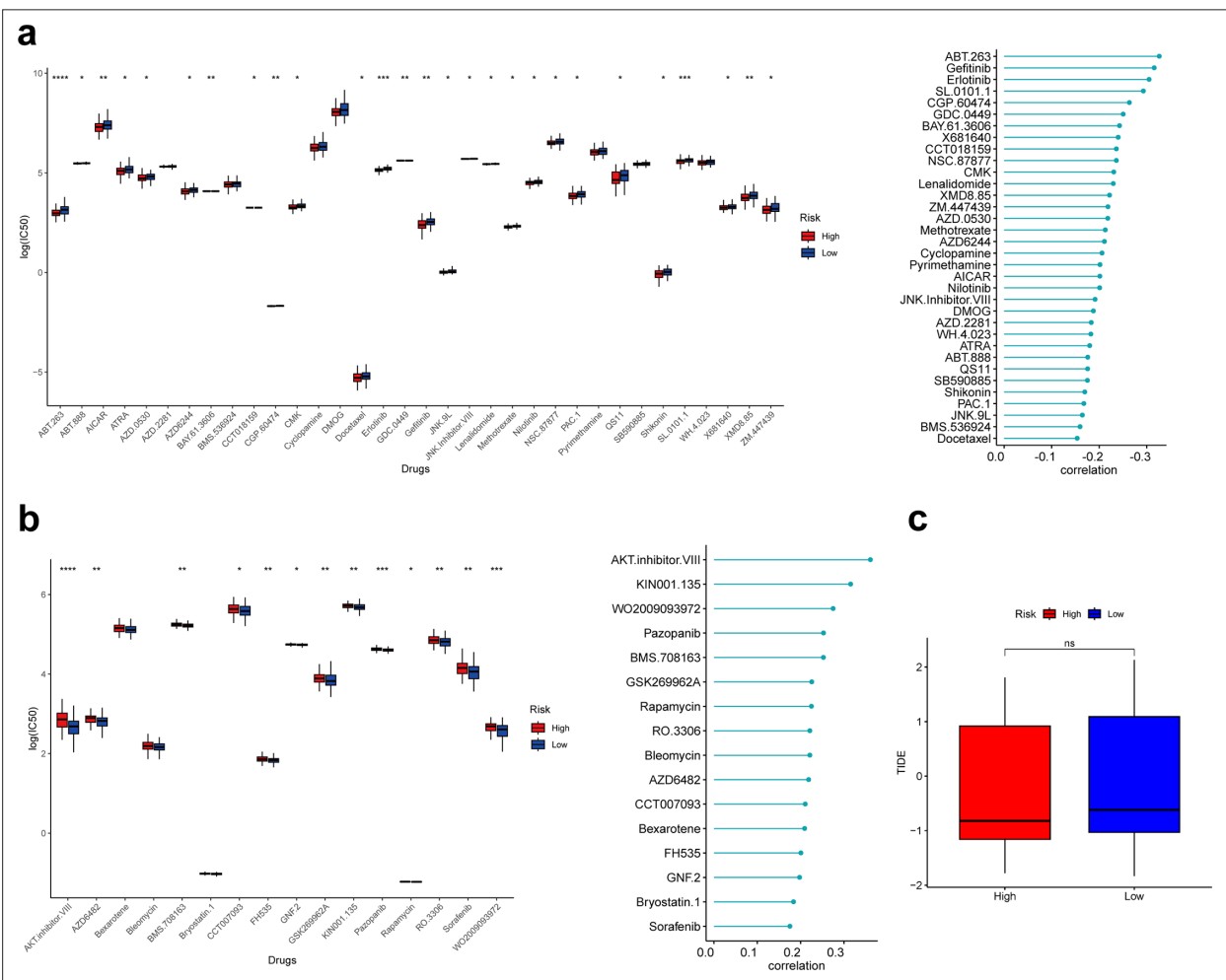

**Figure 7.** Correlation analysis between risk score and drug sensitivity. (**a**) Box plot showing differences of IC$_{50}$ for drugs negatively associated with risk scores in different groups. (**b**) Box plot showing differences of IC$_{50}$ for drugs positively associated with risk scores in different groups. (**c**) Box plot showing differences of TIDE scores in different groups. *p<0.05, **p<0.01, ***p<0.001, ****p<0.0001, ns: p>0.05.

The online version of this article includes the following source code for figure 7:

**Source code 1.** Correlation analysis between risk score and drug sensitivity.

TREM2 reshapes macrophage infiltration, fosters the enrichment and activation of T cells and NK cells, and is correlated with an enhanced response to checkpoint blockade therapy (*Molgora et al., 2020*). In addition, S100A8 and IFI6 were also highly expressed in the M_C2 subpopulation. Studies have reported that a high percentage of S100A8[+] myeloid cell infiltration has been suggested as a potential mechanism for worsening the prognosis of TNBC (*Drews-Elger et al., 2014*; *Rigiracciolo et al., 2022*). Furthermore, IFI6 is an interferon-stimulated gene with anti-apoptotic and metastasis-promoting effects (*Davenport et al., 2023*). These findings demonstrate the heterogeneity of the function of iron death-associated macrophage subpopulations, which may play different roles in the antitumor process. In the original literature, *Prabhakaran et al., 2017* classified breast cancer into nine clusters (Ecotypes) based on single-cell characteristics, which exhibited distinct cellular compositions and clinical outcomes. Among them, Ecotypes-2 is primarily composed of LumA and Normal-like tumors, and patients in this group have the best prognosis, whereas Ecotype-3 is enriched with Basal_SC, Cycling, Luminal_Progenitor, and a Basal bulk PAM50 subtype, and patients in this group have the worst prognosis. In comparison, this study focuses on TNBC, which has a poorer prognosis. In the different distributions of T cell and macrophage subtypes in different TNBC patients, the study results revealed that ferroptosis-related TME in TNBC patients may be valuable for patient survival and outcome prediction. In the future, altering the degree of macrophage or T-cell infiltration in different subpopulations could be potentially valuable in improving the prognosis of TNBC patients.

Intratumor heterogeneity is one of the most important factors contributing to poor clinical outcomes in breast cancer, and ferroptosis is one of the major pathways of activation in highly heterogeneous tumors (*Su et al., 2023*; *Wu et al., 2021*; *Zardavas et al., 2015*). In order to better address the problem of accurately predicting the prognosis of TNBC, we firstly performed the correlation analysis of ferroptosis-related genes with RFS and constructed a risk score model in the publicly available dataset of bulk RNA-seq. TNBC patients were categorized into low- and high-risk groups based on the risk score. In the high-risk group, TNBC patients had higher expression levels of the MIIP, HIST3H2A, CDC25B, TCEB1, HMGCS1, SPC25, TKT, PTTG1, ADA, AK1, and AIMP2 genes. Among them, the cell division cycle 25 (CDC25) family of proteins is dual-specific tyrosine phosphatases responsible including three subtypes, CDC25A, CDC25B, and CDC25C, which are used to regulate cell cycle transitions (*Boutros et al., 2007*; *Cairns et al., 2020*). CDC25B dephosphorylates and activates Cyclin-dependent Kinase 1/Cyclin B (CDK1/Cyclin B), which have been shown to be overexpressed in breast cancer and promote cell cycle progression (*Cairns et al., 2020*; *Guo et al., 2022*). Spindle component 25 (SPC25) is a key component of the nuclear division cycle 80 (NDC80) complex, and its high expression promotes tumor cell proliferation by inducing mitotic disorder (*Yang et al., 2022*). A study showed that SPC25 promotes the proliferation of breast cancer cells, and high levels of SPC25 mRNA levels are associated with high recurrence rates and low survival rates in breast cancer patients (*Wang et al., 2019*). Pituitary tumor transforming gene 1 (PTTG1) has been demonstrated to promote proliferation, migration, and invasion of cancer cells and is a gene that promotes breast cancer development (*Qi et al., 2019*). At the mRNA level, PTTG1 is more present in patients with high histologic grade and with lymph node metastasis, is significantly associated with low patient survival, and is a biomarker suggestive of poor prognosis in breast cancer, which may be related to the regulation of the cell cycle by PTTG1 to promote the development of breast cancer cells (*Meng et al., 2020*). In the low-risk group, TNBC patients had higher expression levels of the TMEM160, EWSR1, BCAT2, PNKP, MLEC, SAP30BP, TBL1XR1, STAG1, NR4A1, TNFRSF9, PSD3, and BAD genes. These genes have different roles in regulating the TME in colorectal cancers, bladder cancers, lymphomas, etc., but they have been rarely reported in TNBC (*Dai et al., 2024*; *Lei et al., 2020*; *Venturutti et al., 2020*). In future studies, these genes are valuable to investigate in the regulation of anti-TNBC.

In TNBC, we found that the risk score model we constructed had independent predictive power for RFS and the predictive performance of this score was stable in an external validation set. Considering the impact of the tumor immune microenvironment on the prognosis of TNBC patients, we further explored the differences in immune cell infiltration in different subgroups of patients. The results showed that T cells CD4 memory-activated, NK cells resting, and monocytes had higher proportions in the high-risk group. This is in line with previous studies, such as that CXCL7 expressed and released by monocytes can stimulate cancer cell migration, invasion, and metastasis (*Wang et al., 2021*). While in the low-risk group, the proportion of NK cells activated and mast cells resting was higher. It has been suggested that the effect of mast cells on tumor invasiveness may vary among different breast

cancer subtypes (*Bense et al., 2017*; *Majorini et al., 2020*). Our findings suggest that infiltration of mast cells resting in the TME is more favorable to the prognosis of patients.

For high-risk TNBC patients, it is crucial to seek more effective drugs for adjuvant therapy in clinical practice. In our study, we found that ABT-263 (navitoclax) and erlotinib exhibited the greatest differences in drug sensitivity among different patient groups. ABT-263 is a mimetic of B-cell lymphoma-2 (BCL-2) homology 3 (BH3) that can increase cellular reactive oxygen species (ROS) levels and induce TNBC cell apoptosis by inhibiting the function of anti-apoptotic proteins (BCL-2, BCL-XL, and BCL-W) (*Lee et al., 2022*; *Oh et al., 2021*). ROS is a key regulatory factor in the occurrence of ferroptosis, and excessive production of ROS can lead to the accumulation of lipid peroxides, ultimately triggering cellular ferroptosis (*Shi et al., 2024*; *Stockwell, 2022*). In addition, erlotinib is an epidermal growth factor receptor (EGFR) tyrosine kinase inhibitor (TKI) that exerts its effects by competitively binding to the ATP site within the catalytic domain and inhibiting the phosphorylation of EGFR, thereby improving the progression-free survival of patients with EGFR mutations (*Roberts et al., 2020*; *Yue et al., 2018*). Studies have shown that β-Elemene can enhance the sensitivity of EGFR-mutated non-small cell lung cancer to erlotinib by upregulating lncRNA H19 and inducing ferroptosis, providing a new approach to overcoming drug resistance in TNBC (*Xu et al., 2023*). EGFR is a receptor tyrosine kinase that promotes the proliferation and invasion of breast cancer by stimulating multiple oncogenic pathways such as Ras-Raf-MEK-ERK, PI3K-AKT-mTOR, and Src-STAT3 (*Wee and Wang, 2017*). The high sensitivity of high-risk patients to erlotinib may be attributed to the higher expression of EGFR in their tumors. Currently, the treatment of TNBC mainly relies on systemic chemotherapy, while small molecule inhibitors and targeted drugs induce cell death through various pathways, holding great promise for antitumor treatment in high-risk patient groups.

This study has some limitations. First, this study is based on the analysis of transcriptomic data, and the integration of multi-omics data is needed to improve the predictive performance of the model. Second, this study is a retrospective analysis of a public dataset, and future in vivo, in vitro experiments as well as prospective clinical trials are needed to further validate the research findings.

In conclusion, this study revealed the TNBC ferroptosis-mediated tumor immune cell clustering based on scRNA-seq data, and the degree of infiltration of different subpopulations of cells had different roles in the prognosis of TNBC patients. On this basis, combined with bulk RNA-seq data, the survival prediction model for TNBC patients was established with excellent predictive performance and stability. The high-risk TNBC patients screened by this prediction model and their sensitive therapeutic drugs can help guide physicians in disease monitoring and precise treatment. The related genes screened also provide important value for TNBC patients to find potential therapeutic targets to improve their prognosis.

# Materials and methods

## Key resources table

| Reagent type (species) or resource | Designation | Source or reference | Identifiers | Additional information |
| --- | --- | --- | --- | --- |
| Software, algorithm | R package Seurat | Microsoft | N/A | Version 4.1.1 |

## Patient data collection and processing

The data used in this study were collected from public datasets. Nine single-cell RNA-seq TNBC samples were obtained from the Gene Expression Omnibus (GEO) database (https://www.ncbi.nlm.nih.gov/geo/query/acc.cgi) under the accession number GSE176078 (*Wu et al., 2021*; *Table 1*).

To construct the model to predict prognosis and immunotherapy efficacy, bulk RNA-seq datasets and clinicopathological information of TNBC were obtained from the GEO database under the accession number GSE25066 (*Hatzis et al., 2011*). The data for external validation were obtained from the GEO database under the accession number GSE86166 (*Prabhakaran et al., 2017*).

## Single-cell RNA-seq data processing

Data were processed using the R package Seurat (v. 4.1.1) for QC. Based on QC metrics suggested in Scanpy tutorial, outlier cells were removed based on relevant feature data (nFeature_RNA, nCount_RNA, percent.mt), that is, cells with less than 200 genes expressed or more than 20% mitochondrial genes counts were filtered out. After quality control, the expression data of each cell was normalized

**Table 1.** Sample information for triple-negative breast cancer in GSE176078.

| Case ID | Gender | Age | T_Stage | Subtype by IHC |
|---------|--------|-----|---------|----------------|
| 3946 | Female | 52 | T2 | TNBC |
| 44041 | Female | 35 | T2 | TNBC |
| 4465 | Female | 54 | T2 | TNBC |
| 4495 | Female | 63 | T1 | TNBC |
| 44971 | Female | 49 | T2 | TNBC |
| 44991 | Female | 47 | NA | TNBC |
| 4513 | Female | 73 | T3 | TNBC |
| 4515 | Female | 67 | T1 | TNBC |
| 4523 | Female | 52 | T2 | TNBC |

IHC, immunohistochemistry; TNBC, triple-negative breast cancer.

separately using the NormalizeData function and the top 2000 highly variable genes (HVGs) of each cell were identified using the FindVariableFeatures function. To remove batch effects among different samples, we utilized the FindIntegrationAnchors function to find anchor genes and then the reclassified datasets were integrated utilized the IntegrateData function. Data normalization was performed using the ScaleData function in Seurat. The RunPCA function was utilized to reduce the dimension of principal component analysis (PCA) for the first 2000 HVGs screened above.

## Genes associated with ferroptosis acquiring

The data of 471 genes (*Supplementary file 2*) associated with ferroptosis were collected from the FerrDb website (http://www.zhounan.org/ferrdb/current/). Then, from the above genes, screened for genes with expression information in the single-cell matrix of TNBC patients. Firstly, based on the clinical data of the samples, the expression values of ferroptosis-related genes in the same type of samples were averaged and presented in a heatmap. Afterward, the expression values of ferroptosis-related genes were averaged across different cell types and also presented in a heatmap.

## Cell subtype identification associated with ferroptosis

Because information of cell type annotation was listed in the original literature of the GSE176078 dataset, the metadata of the original literature was used directly for the next analyses. The R package Seurat (v. 4.1.1) was used to identify cell types and presented results via t-SNE. The NMF is an algorithm based on high-throughput data to identify and cluster out different molecular functional patterns (*Ding et al., 2023*). In this work, subtypes of macrophages and T cells based on genes associated with ferroptosis using the 'NMF' R package (v 0.26).

## Cell–cell communication analysis

Cell–cell communication mediated by ligand–receptor complexes plays an important role in multiple biological processes (*Efremova et al., 2020*). In this study, the 'iTALK' R package was utilized to construct the cellular communication network. First, use the rawParse function to identify the top 50% highly expressed genes for each cell type based on their expression means. Then utilize the FindLR function to identify ligands and receptors among the highly expressed genes. Finally, construct an interaction network based on all the interaction relationships.

## Pseudotime analysis

Pseudotime analysis of scRNA-seq snapshot data helps provide an approximate landscape of gene expression dynamics (*Sugihara et al., 2022*). The 'Monocle'R package (v. 2.28.0) was applied for pseudotime analysis to conduct cellular trajectory. The reduceDimensio function based on the

DDRTree algorithm is used to reduce the dimensions of the data. Based on the Progenitor Cell Biology Consortium database (https://www.synapse.org), the stemness signature was identified via the one-class logistic regression algorithm, then the stemness index of each TNBC cell was calculated by scaling the Spearman correlation coefficients to be between 0 and 1. Eventually, the order Cells function was utilized to sort cells and complete construction of trajectory.

### Transcriptional factor analysis

The Single-cell rEgulatory Network Inference and Clustering (SCENIC) tool enables simultaneous gene regulatory network reconstruction and cell-state identification from scRNA-seq data (*Aibar et al., 2017*; *Van de Sande et al., 2020*). We utilized the 'SCENIC' R package (v.1.3.1) to establish the TFs regulatory network. Specifically, first, co-expression modules are inferred using the 'GENIE3/GRNBOOST' function to identify gene set with co-expressed TFs. Next, the indirect targets are pruned from these modules using cis-regulatory motif discovery (cisTarget). Finally, the AUCell algorithm was utilized to evaluate the activity of regulons.

### Hallmarks gene set enrichment analysis

Hallmarks gene set enrichment analysis was utilized the 'irGSEA' R package (v2.1.5). The ssGSEA algorithm was utilized to conduct differential pathway score between ferroptosis-related immune cell subtypes.

### Construction and validation of the prognostic model

First, based on the RFS data of 178 TNBC patients in the GSE25066 database, this study utilized the 'survival' (v.3.2–7) and 'survminer' (v0.4.8) R package to perform univariate Cox proportional hazards regression analyses on ferroptosis-related genes signature of different immune cell subtypes.

The genes screened for significant association with RFS in the regression analysis ($p<0.01$) were utilized to construct a risk factor-based model by the LASSO method implemented in the 'glmnet' R package (v.4.0–2). TNBC patients were divided into low- and high-risk groups according to the median risk score, and Kaplan–Meier survival curves were utilized to compare the RFS rates between the two groups, $p\text{-value}<0.05$ was considered to be of significance. To validate the performance of the model, receiver operating characteristic curves were demonstrated and the AUC values were calculated for evaluating 3-, 4-, and 5-year RFS rates of TNBC patients in GSE25066 database. In addition, the GSE86166 was utilized as external validation sets to further verify the robustness of the performance of the constructed model in this study.

### Immune cells infiltration and drug sensitivity analysis

Immune cells infiltration is important to the antitumor response, and these cells are diverse among patients (*Iglesias-Escudero et al., 2023*; *Pérez-Romero et al., 2020*). The 'CIBERSORT', 'GSVA', and 'TIMER' R package was utilized to determine the distribution of different immune cell types between low- and high-risk groups. Next, the $IC_{50}$ of 138 chemotherapeutic drugs was calculated for each patient using the 'pRRophetic' R package (v. 0.5), and drugs significantly associated with the risk score were screened. Finally, we calculated the TIDE score (http://tide.dfci.harvard.edu.) for each patient and thus analyzed its difference between low- and high-risk groups.

### Statistical analysis

The R software (v. 4.1.3) was utilized for all data analysis. Wilcoxon rank-sum test was utilized to analyze associations of continuous variables. Log-rank test was utilized to analyze differences in survival curves between groups. p value <0.05 was considered statistically significant.

## Acknowledgements

Our thanks go to the GEO database and the FerrDb website for providing essential data resources. Supported by the National Natural Science Foundation of China (81974268, 82472000, 82304151); Talent Incentive Program, Cancer Hospital Chinese Academy of Medical Sciences (801032247); Cooperation Fund of CHCAMS (CFA202202023); and open project of Beijing Key Laboratory of Tumor Invasion and Metastasis Mechanism, Capital Medical University (2023ZLKF03).

# Additional information

## Funding

| Funder | Grant reference number | Author |
| --- | --- | --- |
| National Natural Science Foundation of China | 81974268 | Yong Wang |
| National Natural Science Foundation of China | 82472000 | Yong Wang |
| National Natural Science Foundation of China | 82304151 | Hao Qin |
| Talent Incentive Program, Cancer Hospital Chinese Academy of Medical Sciences | 801032247 | Yong Wang |
| Cooperation Fund of CHCAMS | CFA202202023 | Hao Qin |
| open project of Beijing Key Laboratory of Tumor Invasion and Metastasis Mechanism, Capital Medical University | 2023ZLKF03 | Hao Qin |

The funders had no role in study design, data collection and interpretation, or the decision to submit the work for publication.

## Author contributions

Xuantong Gong, Conceptualization, Resources, Data curation, Formal analysis, Investigation, Methodology, Writing - original draft, Writing – review and editing; Lishuang Gu, Conceptualization, Resources, Data curation, Investigation, Methodology, Writing - original draft, Writing – review and editing; Di Yang, Data curation, Writing – review and editing; Yu He, Software, Writing – review and editing; Qian Li, Conceptualization, Formal analysis, Supervision, Investigation, Visualization, Writing – review and editing; Hao Qin, Conceptualization, Formal analysis, Funding acquisition, Investigation, Visualization, Writing – review and editing; Yong Wang, Conceptualization, Formal analysis, Supervision, Funding acquisition, Investigation, Methodology, Project administration, Writing – review and editing

## Author ORCIDs

Xuantong Gong  http://orcid.org/0000-0001-8984-5461
Lishuang Gu  https://orcid.org/0000-0003-0746-5620
Di Yang  https://orcid.org/0000-0002-3521-8371
Qian Li  http://orcid.org/0000-0003-0182-3634
Yong Wang  https://orcid.org/0000-0001-7682-0433

## Ethics

This study used publicly available, anonymized datasets (GSE176078, GSE25066, GSE86166) approved by original ethics committees. Per repository agreements, no additional ethics approval or consent was needed for secondary analysis of de-identified data.

Reviewer #2 (Public review): https://doi.org/10.7554/eLife.100923.3.sa1
Author response https://doi.org/10.7554/eLife.100923.3.sa2

# Additional files

## Supplementary files

Supplementary file 1. Univariate Cox regression analysis of ferroptosis-related genes and disease-free survival in patients with triple-negative breast cancer.

Supplementary file 2. Ferroptosis-related genes.

MDAR checklist

## Data availability

The current manuscript is a computational study, so no data have been generated for this manuscript. Modelling code is uploaded as figure level Source code.

The following previously published datasets were used:

| Author(s) | Year | Dataset title | Dataset URL | Database and Identifier |
|-----------|------|---------------|-------------|-------------------------|
| Al-Eryani G, Roden DL, Junankar S, SZ Wu | 2021 | A single-cell and spatially resolved atlas of human breast cancers | https://www.ncbi.nlm.nih.gov/geo/query/acc.cgi?acc=GSE176078 | NCBI Gene Expression Omnibus, GSE176078 |
| Hatzis C, Pusztai L, Valero V, Booser DJ | 2011 | Genomic predictor of response and survival following neoadjuvant taxane-anthracycline chemotherapy in breast cancer | https://www.ncbi.nlm.nih.gov/geo/query/acc.cgi?acc=GSE25066 | NCBI Gene Expression Omnibus, GSE25066 |
| Prabhakaran S, Rizk VT, Ma Z, Cheng CH | 2017 | Evaluation of invasive breast cancer samples using a 12-chemokine gene expression score: correlation with clinical outcomes | https://www.ncbi.nlm.nih.gov/geo/query/acc.cgi?acc=GSE86166 | NCBI Gene Expression Omnibus, GSE86166 |

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
