## [Editor Report · eLife Assessment]

This study presents a **useful** finding for the ferroptosis-mediated tumor microenvironment (TME) in triple-negative breast cancer (TNBC) using public single-cell RNA sequencing (scRNA-seq) and bulk RNA sequencing data. The data were collected and analyzed using **solid** and validated methodology and can be used as a starting point for functional studies of TME in TNBC. The work will be of interest to medical biologists working in the field of TNBC.

---

## [Referee Report · Reviewer #2 (Public review)]

Summary:

This study aims to explore the ferroptosis-related immune landscape of TNBC through the integration of single-cell and bulk RNA sequencing data, followed by the development of a risk prediction model for prognosis and drug response. The authors identified key subpopulations of immune cells within the TME, particularly focusing on T cells and macrophages. Using machine learning algorithms, the authors constructed a ferroptosis-related gene risk score that accurately predicts survival and the potential response to specific drugs in TNBC patients.

Strengths:

The study identifies distinct subpopulations of T cells and macrophages with differential expression of ferroptosis-related genes. The clustering of these subpopulations and their correlation with patient prognosis is highly insightful, especially the identification of the TREM2+ and FOLR2+ macrophage subtypes, which are linked to either favorable or poor prognoses. The risk model thus holds potential not only for prognosis but also for guiding treatment selection in personalized oncology.

---

## [Author Response]

The following is the authors’ response to the original reviews

**Public Reviews:**

**Reviewer #1 (Public review):**
Summary:Triple-negative breast cancer (TNBC) accounts for approximately 15-20% of all breast cancers. Compared to other types of breast cancer, TNBC exhibits highly aggressive clinical characteristics, a greater likelihood of metastasis, poorer clinical outcomes, and lower survival rates. Immunotherapy is an important treatment option for TNBC, but there is significant heterogeneity in treatment response. Therefore, it is crucial to accurately identify immunosuppressive patients before treatment and actively seek more effective therapeutic approaches for TNBC patients.Strengths:In this work, the authors collected and integrated data from single cells and large volumes of RNA sequencing and RNA-SEQ to analyze the TME landscape mediated by genes associated with iron death. On this basis, the prediction model of prognosis and treatment response of 131 patients was constructed using a machine learning algorithm, which is beneficial to provide individualized and precise treatment guidance for breast cancer patients.

Thank you for your appreciation of our work. We are encouraged by your positive feedback and will continue to explore new avenues in personalized medicine for breast cancer.

Weaknesses:However, there are still some issues that need to be clarified:(1) The description of the research background is too brief and concise, and it is necessary to add some information about the limitations of existing methods and the differences and advantages of this study compared with other published relevant studies, so as to better highlight the necessity and research value of this study.

Thank you for your suggestions. We have supplemented the research background and compared the differences between this study and other studies, further highlighting the research value of our study.

(2) This study is a retrospective analysis of a public data set and lacks experimental validation and prospective experiments to support the results of bioinformatics analysis. This should be added to the acknowledgment of limitations in the study.

Thank you for the constructive feedback. We also acknowledge that the lack of experimental evidence is one of the limitations of this study. Therefore, we plan to conduct in vivo and in vitro experiments in our future research to support the findings of our bioinformatics analysis, and have already supplemented the relevant content in the limitations of Discussion.

**Reviewer #2 (Public review):**
Summary:This study aims to explore the ferroptosis-related immune landscape of TNBC through the integration of single-cell and bulk RNA sequencing data, followed by the development of a risk prediction model for prognosis and drug response. The authors identified key subpopulations of immune cells within the TME, particularly focusing on T cells and macrophages. Using machine learning algorithms, the authors constructed a ferroptosis-related gene risk score that accurately predicts survival and the potential response to specific drugs in TNBC patients.Strengths:The study identifies distinct subpopulations of T cells and macrophages with differential expression of ferroptosis-related genes. The clustering of these subpopulations and their correlation with patient prognosis is highly insightful, especially the identification of the TREM2+ and FOLR2+ macrophage subtypes, which are linked to either favorable or poor prognoses. The risk model thus holds potential not only for prognosis but also for guiding treatment selection in personalized oncology.

Thank you for your thorough review and insightful comments.

Weaknesses:The study has a relatively small sample size, with only 9 samples analyzed by scRNA-seq. Given the typically high heterogeneity of the tumor microenvironment (TME) in cancer patients, this may affect the accuracy of the conclusions. The scRNA-seq analysis focuses on the expression of ferroptosis-related genes in various cells within the TME. In contrast, bulk RNA sequencing uses data from tumor samples, and the results between the two analyses are not consistent. The bulk RNA sequencing results may not accurately capture the changes happening in the microenvironment.

Thank you for your constructive feedback. Although this study only included 9 samples, given the limited availability of scRNA-seq datasets for untreated TNBC in public databases, we chose to utilize a dataset that contains a relatively larger number of untreated TNBC samples. We are fully aware of the complexity and high heterogeneity of the TME. Despite the limited sample size, we first conducted rigorous quality control on the data and, based on this, preliminarily revealed the landscape of the TME mediated by ferroptosis-related genes. These findings provide a new perspective for understanding the biological mechanisms underlying the onset and progression of breast cancer. To enhance the reliability and generalizability of our research results, we plan to strive to expand the sample size in future work and consider integrating other omics technologies, such as proteomics and metabolomics, with scRNA-seq data for a more in-depth exploration of the complex interactions within the TME.

We also agree with your viewpoint that scRNA-seq data reveals gene expression within individual cells, while bulk RNA-seq data reveals the average gene expression in tumor tissues, and there are differences in data acquisition and processing methods between the two. However, we believe that there are also some close connections between them in terms of gene expression levels. By comparing the expression specificity of marker genes for specific cell types in breast cancer tissues, we found that they are correlated with patient prognosis, and the results have been validated in both internal and external validation sets. Thank you once again for your valuable suggestions, which will play an important guiding role in our subsequent research.

**Reviewer #1 (Recommendations for the authors):**
(1) The breast cancer scRNA-seq dataset files of GSE176078 include 10 TNBC primary tumors (DOI:10.1016/j.compbiomed.2023.107066). However, in this study, only 9 cases were listed, please explain the reason for the data exclusion.

Thank you for your questions. Although it was clearly stated in the original paper that "To elucidate the cellular architecture of breast cancers, we analyzed 26 primary pre-treatment tumors, including 11 ER+, 5 HER2+ and 10 TNBCs, by scRNA-Seq (Supplementary Table 1)," upon downloading and carefully examining the patient information in Supplementary Table 1, we only included 9 patients explicitly labeled as TNBC in our study (https://pmc.ncbi.nlm.nih.gov/articles/PMC9044823/#SD1).

(2) The description of the technique in the methods section should be more detailed, such as parameter settings, quality control standards, etc.

Thank you for your valuable suggestions. We have already supplemented the relevant content in the methods section.

(3) Please check and correct formatting errors to improve readability, such as lines 176 and 177.

We were really sorry for our careless mistakes. Thank you for your reminder. We have corrected the “Pseudotime analysis with scRNA-seq data helps to obtain an approximate landscape of gene expression dynamics” into “Pseudotime analysis of scRNA-seq snapshot data helps to provide an approximate landscape of gene expression dynamics”. And we have further checked and revised the formatting errors of the manuscript.

**Reviewer #2 (Recommendations for the authors):**
(1) In multiple sections of the paper, abbreviations are used without being defined when first mentioned.

We were really sorry for our careless mistakes. Thank you for your reminder. We have already added definitions for the abbreviations in both the abstract and the main text.

(2) The authors should analyze whether the transcription factors in Figure 2 are correlated with the expression of ferroptosis-related genes.

Thank you for your valuable feedback. Some transcription factors in Figure 2 correlate with the expression of ferroptosis-related genes, which we have supplemented in the Discussion.

(3) Figures 3d and 4e lack explanations for the axis values, and for Figure 4e, is the unit of the y-axis labeled "survival" in days?

Thank you for your valuable feedback. We apologize for the lack of explanations for the axis values in Figures 3d and 4e and we have made revisions to both figures accordingly. We have noted that the unit "survival" on the y-axis of Figure 4e is in years, and we have already made the necessary supplement to clarify this. Thank you very much for your reminder.

(4) The authors conducted their analysis using public databases but did not cite the original literature, nor did they discuss the similarities and differences between their findings and those in the original studies.

Thank you for your valuable suggestions, and we deeply apologize for our carelessness. We have supplemented the original literature in the references and discussed the differences between this study and the original literature in the Discussion.

(5) Some figures, particularly those involving heatmaps and t-SNE plots (e.g., Figures 1 and 3), present dense and complex data that may be challenging for readers to interpret. The heatmaps (Figure 1e-f and 3d) include many genes, but it is unclear how these genes were selected, and the scale of gene expression differences is difficult to interpret. Simplifying these figures by focusing on the most differentially expressed and clinically relevant genes (e.g., those with prognostic value) would improve readability.

Thank you for your valuable suggestions. The t-SNE plots in Figures 1 and 3 primarily serve as a dimensionality reduction technique to visually present the clustering of multiple cells or samples based on gene expression, aiding readers in quickly identifying cell subpopulations. The heatmaps, on the other hand, are mainly used to showcase the differential expression of ferroptosis-related genes across different clinicopathological classifications and cell subpopulations, with varying shades of color helping readers quickly recognize gene expression differences among different cell subpopulations. The genes included in the heatmaps (Figures 1e-f and 3d) are sourced from the FerrDb website. We have uploaded the list of ferroptosis-related genes used in this study as Supplementary Table 1 and added the relevant steps in Method 2.3.

(6) The study analyzes the expression of ferroptosis-related genes in different immune cells within the TME. The authors should discuss how these changes in gene expression may impact the function and behavior of immune cells.

Thank you for your valuable feedback. We have supplemented the discussion with detailed effects of the main differential genes (FOLR2 and TREM2) on the tumor immune response.

(7) The authors analyzed the expression of ferroptosis-related genes in immune cells using single-cell sequencing data. However, they subsequently applied the selected genes to perform a risk factor analysis in tumor cells. Is the expression and function of these genes the same in immune cells and tumor cells? This seems questionable.

Thank you very much for your suggestion. We also believe that there may be differences in the expression and function of genes between immune cells and tumor cells. However, some genes may exhibit similarities in their expression and function in immune cells and tumor cells, especially within the tumor immune microenvironment, due to the complex and tight interactions between immune cells and tumor cells (as shown in Figures 1d and 2h), and their expression levels can be related to the onset, progression, and prognosis of tumors.

(8) While the risk score model based on ferroptosis-related genes is promising, it lacks experimental validation, which weakens the strength of the conclusions. The authors should consider conducting in vitro or in vivo experiments. These functional studies would provide essential evidence to support the model's predictive capability.

Thank you for the constructive feedback. We fully recognize the importance of conducting functional studies to substantiate the predictive capability of the model. Therefore, we plan to conduct in vitro and in vivo experiments in our future research to provide the necessary evidence and further validate the model's effectiveness.

(9) The manuscript predicts sensitivity to 27 drugs based on the risk score, but it lacks mechanistic insight into why patients in the high-risk group might be more responsive to certain drugs. Including a more detailed discussion of the molecular mechanisms underlying this drug sensitivity, particularly linking ferroptosis-related genes to drug metabolism or efficacy, would provide a stronger rationale for the clinical application of these findings.

Thank you very much for your valuable suggestions. In the discussion, we thoroughly analyzed the mechanism of action of the drugs (ABT-263 and erlotinib) with the greatest difference in sensitivity between high-risk and low-risk groups, as well as their correlation with ferroptosis.